# Hardware Aware Training for Efficient Keyword Spotting on General Purpose and Specialized Hardware

anonymous

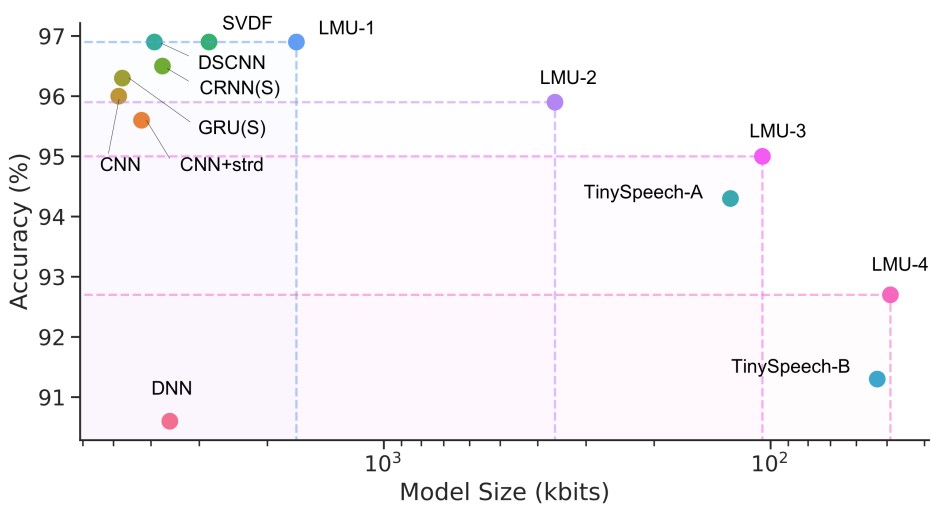

**Figure 1: Scatter plot of the model size and accuracy data in Table 1. Model size is shown on an inverted log scale (right is better). The LMU models are consistently smaller and more accurate over the model space, shown by their being in the top right. NB: TinySpeech models are stateless (state is reset between inferences) and non-streamable, hence not appropriate for real-time deployment. Other more recent models of this type are discussed in the text but not included in this figure.**

## ABSTRACT

Keyword spotting (KWS) provides a critical user interface for many mobile and edge applications, including phones, wearables, and cars. As KWS systems are typically 'always on', maximizing both accuracy and power efficiency are central to their utility. In this work we use hardware aware training (HAT) to build new KWS neural networks based on the Legendre Memory Unit (LMU) that achieve state-of-the-art (SotA) accuracy and low parameter counts. This allows the neural network to run efficiently on standard hardware ($212\,\mu W$). We also characterize the power requirements of custom designed accelerator hardware that achieves SotA power efficiency of $8.79\,\mu W$, beating general purpose low power hardware (a microcontroller) by 24x and special purpose ASICs by 16x.

**ACM Reference Format:**

anonymous. 2020. Hardware Aware Training for Efficient Keyword Spotting on General Purpose and Specialized Hardware. In *Proceedings of ACM Conference (Conference'17)*. ACM, New York, NY, USA, 5 pages. https://doi.org/10.1145/nnnnnnn.nnnnnnn

**Keywords**  speech processing, keyword spotting, on-device inference, online inference, keyword spotting hardware, edge AI, low power, deep learning accelerator, TinyML, hardware aware training

## 1 INTRODUCTION

There are a wide variety of keyword spotting deep neural networks available, including those based on CNNs, LSTMs, GRUs, and many variants of these. However, commercially viable networks have several constraints often ignored by research focused efforts. In this more constrained setting, neural networks must be:

(1) *Stateful*: The network cannot assume to know when a keyword is about to be presented. As a result, the starting state of the network cannot be known in advance, but is determined by whatever processing has happened recently – not by being reset to a known 'zero' state.

(2) *Online* (or 'streaming'): The most responsive, low-latency networks will process audio data as soon as it is available and in real-time. Many methods are often tested on the assumption that large windows of data are available all at once. However, at deployment, waiting for large amounts of data introduces undesirable latencies. As well, reusing previously processed data, as done by RNNs, can lead to efficiency gains.

(3) *Quantized*: Quantization to 8-bit weights and activities is becoming standard for mobile or 'edge' applications. Quantization allows more efficient deployment on low power, edge hardware.

(4) *Power efficient*: While quantization helps with power efficiency, it is not the sole determiner of the power required by a network. For instance, the number and type of computations performed are also important. Specific focus on the power efficiency of the network, and its viability for deployment on available hardware is critical for commercial applications.

In this paper, we use a method of hardware aware training (HAT) that directly trains a network for efficient hardware deployment, accounting for hardware assumptions during model development. This provides a practical method for meeting such constraints.

As a result, we focus on comparing this work to recent SotA results that share interest in these constraints. All of the new results we report also satisfy these constraints. As a consequence, our main metrics of interest will be: accuracy; size of the model (in bits; the number of parameters times the bits per parameter); and power usage in a real-time setting. In what follows we describe new optimization, algorithmic, and hardware techniques that have allowed us to develop a highly power efficient KWS algorithm and hardware platform. Critically, we demonstrate that these same methods can be used to target different hardware platforms (both general and special purpose). To the best of our knowledge, we present better than current SotA results on each of these metrics and for each platform.

## 2 THE LEGENDRE MEMORY UNIT

The recurrent neural network (RNN) that lies at the heart of our algorithm is called the Legendre Memory Unit (LMU), which we have recently proposed [13].[1] The LMU consists of a linear 'memory layer' and a nonlinear 'output layer' that are recurrently coupled both to themselves and each other. A distinguishing feature of the LMU is that the linear memory layer is optimal for compressing an input time series over time. The output of this layer represents the weighting of a Legendre basis, which gives rise to the LMU's name.

Because of this provable optimality, unlike past RNNs (including LSTMs, GRUs, and so on) the LMU has fixed recurrent and input weights on the linear layer. As well, the theoretical characterization of the LMU permits intermediate representations to be decoded, providing a degree of explainability to the functioning of the network.

In the original LMU paper, it was shown that on a task requiring the memory of a time-varying signal, the LMU outperforms the LSTM with a $10^6$x reduction in error, while encoding $10^2$ more timesteps, and using 500 versus 41,000 parameters. In some ways this is not surprising, as the LMU is optimized for this task. Nevertheless, the LMU also outperforms all previous RNNs on the standard psMNIST benchmark task by achieving 97.15% test accuracy, compared to the next best network (dilated RNN) at 96.1% and

---

[1]The LMU is a patent pending technology of Applied Brain Research Inc., free for academic research, educational and personal uses. Please contact ABR for commercial use licensing at info@AppliedBrainResearch.com

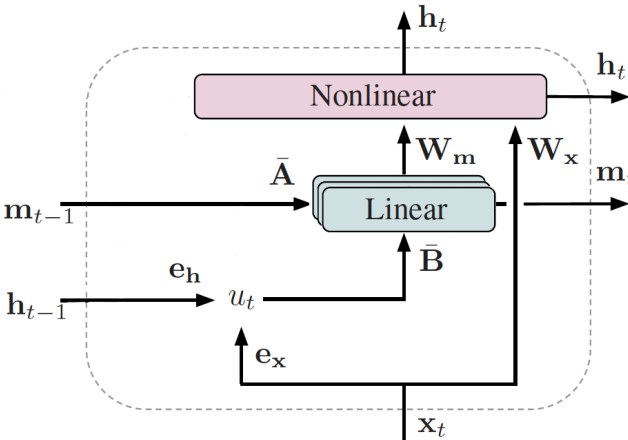

**Figure 2: The LMU architecture used in this work. This differs from the original LMU [13], in that there are multiple linear layers and fewer connections (see text for details).**

the LSTM at 89.86%. Again, the LMU used far fewer parameters ~102,000 versus ~165,000 (a reduction of 38%).

Because the LMU is designed to be optimal at remembering information over a window, while receiving streamed input, and because it also tends to use fewer parameters while achieving high accuracy, it is well-suited to the constraints of real-world KWS tasks.

### 2.1 Model architecture

In this work, we have modified the originally proposed LMU in a number of ways (see Figure 2). In particular, we have removed the connection from the nonlinear to the linear layer, the connection from the linear layer to the intermediate input, and the recurrent connection from the nonlinear layer to itself. As well, we have included multiple linear memory layers in the architecture. We found that this was important for improving performance on the KWS task.

The resulting architecture, depicted in Figure 2, is thus described by the following equations:

$$h_t = f\left(\mathbf{W_x x}_t + \mathbf{W_m m}_t + \mathbf{b}\right)$$
$$u_t = \mathbf{e_x}^\mathsf{T} \mathbf{x}_t + \mathbf{e_h}^\mathsf{T} \mathbf{h}_{t-1}$$
$$\mathbf{m}_t = \bar{\mathbf{A}} \mathbf{m}_{t-1} + \bar{\mathbf{B}} u_t$$

where each of the variables is defined as depicted in Figure 2, and the nonlinearity we use for this application is the ReLU.

We refer to this architecture as a single LMU layer. The final network we test includes multiple LMU layers and a feedforward output layer.

## 3 RESULTS

### 3.1 Dataset, methods, and metrics

Our methods and metrics follow standard practices for the Speech-Commands dataset (see, e.g., Rybakov et al. [11], Warden [15]). As specified in Warden [15] we split the data into training, validation,

and testing sets with one second speech samples at 16 kHz. The network is trained on twelve labels: the ten keywords, plus silence and unknown tokens. All accuracy results are on the test data only (see Table 1 and Figure 1).

The methods we use to build the LMU models all leverage hardware aware training (HAT). This extends standard quantization aware training to precisely match the hardware on which the models will be deployed. This means that all model elements are matched to the bit precisions assumed throughout a design. Quantization aware training typically makes assumptions not satisfied by hardware.[2] As a result the reported accuracies for the LMU models are the expected, real-world, deployed accuracies.

All LMU models and the results from Rybakov et al. [11] are for stateful, quantized and online KWS applications. The results from Wong et al. [17] are quantized, but their latency and statefulness is not reported. Because the amount of quantization is different between different models, we have measured the model size in kilobits (kbits) instead of parameter count. The kbits are the number of parameters multiplied by the number of bits per parameter to give a consistent model size metric.

In Table 1 we show the results from four different LMU models. The first model (LMU-1) uses 8-bit weights, while the remaining three models use 4-bit weights. All LMU models use 7-bit activations. LMU-1 and LMU-2 are not pruned. LMU-3 has 80% pruning performed and LMU-4 has 91% of its weights pruned.

## 3.2 Comparison to other work

We compare our results to Google's latest KWS paper [11], updated in July of 2020, ARM's recent results [2] and DarwinAI's announcement from August of 2020 [17] and October of 2020 [18]. As shown in Table 1, the LMU models outperform Google results and Darwin's first announcement in terms of accuracy and size. For instance, LMU-1 is the same accuracy as the best Google model, while using 41% fewer bits. As well, LMU-2 is comparable in accuracy to the CNN [11], while using 11.7x fewer bits in the final model. In comparison to the generally smaller models of Wong et al. [17], the LMUs show significant accuracy improvements. Specifically, the LMU-3 reduces the error by 14% relative to TinySpeech-A, while using 17% fewer bits. Similarly, the LMU-4 reduces error by 19% relative to TinySpeech-B while using 8% fewer bits. Similarly, compared to recent work from ARM [2], the LMU-2 is outperforming all of the tested networks, while being small enough to fit on the smallest processor tested (an ARM M4F).

However, recent work reported in Li et al. [9] and Wong et al. [18] describe convolution approaches that are highly competitive. Specifically the largest TENet model (800 kbits) achieves 96.6%,[3] while the smallest (136 kbits) achieves 96.0%. Similarly, in [18] for smaller networks, TinySpeech-Y (49 kbits) achieves 93.6% accuracy and TinySpeech-X (86 kbits) achieves 94.6%. Critically, neither of these methods are stateful (i.e. network state is reset between inferences), which is known to boost accuracy, both have a minimum 1s latency (as convolutions are done on the entire 1s samples), and

---

[2]For instance, activities are often asymmetrically quantized to unsigned 8 bits, but in a hardware implementation 7-bit quantization is more appropriate since one bit is required for a signed two's complement representation to allow 8-bit multiplication with weights.
[3]We have also generated a 720kbit network at 96.5%, not included in Table 1.

**Table 1: Recent KWS accuracy results with model sizes.**

| Model | Accuracy (%) | Model Size (kbits) | Reference |
|---|---|---|---|
| DNN | 90.6 | 3576 | Rybakov et al. [11] |
| CNN+strd | 95.6 | 4232 | |
| CNN | 96.0 | 4848 | |
| GRU (S) | 96.3 | 4744 | |
| CRNN (S) | 96.5 | 3736 | |
| SVDF | 96.9 | 2832 | |
| DSCNN | 96.9 | 3920 | |
| TinySpeech-A | 94.3 | 127 | Wong et al. [17] |
| TinySpeech-B | 91.3 | 53 | |
| LMU-1 | 96.9 | 1683 | This work |
| LMU-2 | 95.9 | 361 | |
| LMU-3 | 95.0 | 105 | |
| LMU-4 | 92.7 | 49 | |

both are not streamable. Consequently these networks are not appropriate for real-time deployment, and reported results are not reflective of real-world performance. As such we mention them for completeness, not as an appropriate comparison.

As noted in Section 3.1, the LMU models are all stateful, streamable, and developed with HAT (as are those from Google and ARM). Hence the reported accuracies can be realized on hardware in real-time, real-world applications.

## 4 POWER USAGE ON GENERAL PURPOSE AND SPECIALIZED HARDWARE

### 4.1 Power modeling and results

While power use will scale with model size on standard hardware, suggesting the LMU models will be very efficient, an even more efficient implementation can be obtained using custom designed hardware. Hence, we have designed low power digital hardware to natively implement the necessary computations for the LMU models discussed in Section 3. Here we report results on power modeling of the LMU-2 architecture, which strikes a balance between small size and high accuracy. The design is flexible, allowing for different degrees of parallelism, depending on the speed, power, and area requirements. We considered a variety of designs across different clock frequencies, while always ensuring that the timing constraints of the SpeechCommands models proposed above (40 ms windows updated every 20 ms) are satisfied in real time.

To estimate the power of our design, we established cycle-accurate power envelopes of our design using ABR's proprietary, silicon-aware, hardware-software co-design mapping tools. Total power usage is determined with these envelopes using publicly available power data [3, 7, 19]. Multiply-accumulate (MAC) and SRAM dynamic and static power, are the dominant power consumers in the design. We also included dynamic power estimates for multipliers, dividers, and other components as a function of the number of transistors in the component, and the power cost per transistor of the MAC. All estimates are for a 22nm process.

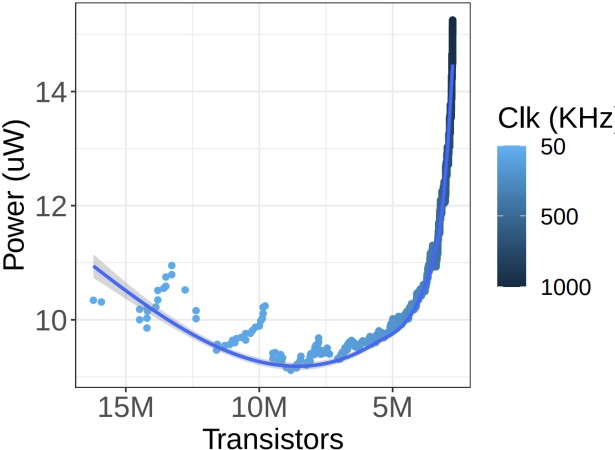

**Figure 3: Power and area trade-off for different clock frequencies of our custom hardware design. Blue dots indicate specific designs considered while varying the number of components and the clock's frequency.**

To estimate the number of transistors, and hence the area, of the design we generated RTL designs of each of the relevant components, and used the yosys open source tool [16] and libraries to estimate the number of transistors required for the total number of components included in our network.

Figure 3 shows the resulting power/area trade-off for our LMU-based design. As can be seen, the lowest power design we found sits at 8.79 $\mu$W (92 kHz clock) and 8,052,298 transistors. For this design, the throughput for one 20 ms frame is 13.38 ms and the latency for the 40 ms update is 39.59 ms, meaning the design runs in real time. Note that all designs depicted in Figure 3 are real-time.

## 4.2 Comparisons across hardware implementations

There have been several recent results published noting low power specialized hardware for keyword spotting. Those we were able to find that had similar or lower numbers are not of comparable complexity or accuracy to the networks we describe here. For instance, Wang et al. [14] claim sub-300 nW power, but only detect a single keyword. Similarly, Giraldo and Verhelst [4] claim less than 5 $\mu$W, but only detect 4 keywords and report accuracy in the low 90s. In contrast, Giraldo et al. [5] uses the SpeechCommands, but the accuracy is 90.9% for 10.6 $\mu$W. A similar result is reported by Shan et al. [12] who achieve 90.8% on this dataset for 16.11 $\mu$W. A main distinguishing feature of our result above is the high accuracy, which is usually very difficult to achieve in a power constrained setting. Our use of the LMU and HAT combine to provide SotA performance.

Because we use HAT, it is straightforward to run the LMU networks on different hardware and compare across them. In this section we compare an implementation on an off-the-shelf ARM M4F, on an 'idealized' ARM M4F, on our hardware design from the previous section, and on the Syntiant NDP10x special purpose keyword spotting chips (see Table 2).

**Table 2: Summary of hardware power results for Speech-Commands keyword spotting.**

| Hardware | Model | Accuracy (%) | Power ($\mu$W) |
|---|---|---|---|
| ARM M4F | LMU-2 | 95.9 | 212 |
| ARM M4F (Optimal) | LMU-2 | 95.9 | 119 |
| Syntiant NPD10x | – | 94.0 | 170 |
| This work | LMU-2 | 95.9 | 9 |

We implemented the LMU keyword spotter on an ARM M4F clocked at 120 MHz, which processes 1 s of audio in 143,678 $\mu$s (0.14 s). This means that 17.24 million cycles are used to process one second of audio. The lowest power setting of the ARM M4 is rated at 12.26 $\mu$W/MHz on the ARM M4 datasheet [1], which results in 212 $\mu$W of power for this model. Thus our design from Section 4.1 is 24x more power efficient. A recent world-record efficiency was reported by Racyics and GlobalFoundries with a power efficiency of 6.88 $\mu$W/MHz [8] for an ARM M4F. Using that idealized power efficiency, the ARM M4F would use 119 $\mu$W of power. This suggests that our design is 14x more power efficient than running on state-of-the-art low power general purpose hardware.

Holleman [6] reports energy per frame on the SpeechCommands dataset for the Syntiant NDP10x special purpose chip at 3.4 $\mu$J. For real-time computation with a standard window stride of 20ms, the network needs to process 50 frames per second, well within the inference time of 10 ms of the chip. This rate of processing results in a power usage of 170 $\mu$W. Syntiant has also reported a power usage of 140 $\mu$W [10]. As well, the network achieves an accuracy of 94%, with a network size of 4456 kbits (assuming 8-bit weights, which is not reported). As a result, our network is more accurate with 95.9% accuracy and is 16-19x more power efficient with our hardware design than the Syntiant special purpose hardware.

Finally, we note that because the LMU is parameter efficient, with 12x fewer parameters than the Syntiant network, it potentially eliminates the need for special purpose hardware. Specifically, the LMU-2 running on the M4F uses (212 $\mu$W), compared to Syntiant's special purpose hardware at (170 $\mu$W). This suggests that the LMU-3 (with one third the parameters of LMU-2) will run for less power, while still achieving higher accuracy.

## 5 CONCLUSION

LMU-based keyword spotting networks are highly efficient, surpassing recent state-of-the-art results from Google, ARM, and Syntiant, in terms of accuracy, size, and power efficiency over a wide range. These improvements become more pronounced with special purpose-designed hardware resulting in >14x reduction in power use compared to current state-of-the-art offerings. Notably, these conclusions are drawn in the context of real-world, real-time deployment of keyword spotting systems.

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
