# OpenReview forum: "Hardware Aware Training for Efficient Keyword Spotting on General Purpose and Specialized Hardware"
_tinyml.org/tinyML/2021/Research_Symposium — tinyML 2021 Regular_

### Official Review · AnonReviewer1 · 2021-01-26

**Overall Merit Score:** 1

**Brief Summary:**

Proposes:

1.) a new network cell (LMU) for keyword spotting networks

2.) a hardware-aware training technique

3.) results of achievable accuracy-complexity performance using 1.) and 2.)

4.) a HW and micro-processor mapping of the above

**Detailed Comments:**

The paper needs improvement regarding:

Providing more details on everything, such that it is clear what has exactly been done / assumed:
- discussion of "modified LMU", but not clear what input / layer dimensionality is used.
- pruning technique mentioned but not explained
- hardware-aware training: not clear how HW is exactly modeled
- chip/hardware architectures leading to figure 3 not described

Provide a figure with the original LMU vs the modified one.

Give a table with all assumed LMU dimensions (nb of linear layers, their sizes,...)

Is the input data preprocessed (E.f. MFCC extraction) or raw data?

"LMU outperforms the LSTM with a 10^6x reduction in error, while encoding 102 more timesteps, and using 500 versus 41,000 parameters" ==> We normally do not express errors as ...x smaller. Also weird to get something a milion times more accurate with only 500 parameters?

How are the outputs of the multiple linear layers combined?

What is the "intermediate" input?

What hardware architecture template s used? What spatial and temporal unrolling? How many MACs were in the most optimal design? What memory hierarchy was used? Separate I and W and O memories, or joint?  Etc...

How is the power consumption exactly estimated. The "proprietary in house tool" is not well explained. Does this uses actual switching activity files? Extracted parasitics, ...?

**Paper Strengths:**

1.) timely and relevant topic

2.) claimed performances are good and very competitive

3.) cross-layer optimization and HW-algorithm co-optimization is important.

**Paper Weaknesses:**

Not enough details / information in the paper to learn / understand what has actually been done. Impossible to justify and/or even judge the claims made.

e.g.:

    - discussion of "modified LMU", but not clear what input / layer dimensionality is used.
    - pruning technique mentioned but not explained
    - hardware-aware training: not clear how HW is exactly modeled
    - chip/hardware architectures leading to figure 3 not described

The paper feels more like a commercial document, and not like a research paper.

**Poster (If Paper Is Rejected):**

1: No, paper is below bar for poster as well

**Reviewer Confidence:**

5: The reviewer is absolutely certain that the evaluation is correct and very familiar with the relevant literature

---

### Official Review · AnonReviewer2 · 2021-01-27

**Overall Merit Score:** 4

**Brief Summary:**

The paper presents a keyword spotting model, based on Legendre Memory Unit (LMU). The model is mapped on ARM M4F clocked at 120 MHz, and the paper presents the performance, accuracy, and power dissipation results. It also includes the estimated power dissipation for potential ASIC implementation. It also provides a short survey in the recent state of the arts along with the comparison between the proposed work and the prior arts.


**Detailed Comments:**

The paper is quite inclusive. It is about a new resource-efficient model for keyword spotting, which is well evaluated and compared to the prior arts.



**Paper Strengths:**

The proposed design, LMU, is interesting, and indeed seems to be effective in terms of accuracy, throughput, online, model size, and power consumption.


**Paper Weaknesses:**

No major weakness. Some minor issues are:
The paper needs some formatting issues. e.g., Fig. 1 needs to be downsized and placed somewhere else, not in the first of the first page.
The paper is not fully blind. It refers to a paper from the same group [14].


**Poster (If Paper Is Rejected):**

1: Yes, ok for poster sesion to nurture work

**Reviewer Confidence:**

5: The reviewer is absolutely certain that the evaluation is correct and very familiar with the relevant literature

---

### Official Review · AnonReviewer4 · 2021-01-27

**Overall Merit Score:** 4

**Brief Summary:**

This paper presents an extension to the Legendre Memory Unit (LMU) along with proposed implementations and their energy efficiencies.
The paper claims very high accuracy results at a fraction of the parameter count that known method use.
The LMU extensions are quantized (down to 8 or 4bits) and are mappable to both software and hardware.


**Detailed Comments:**

RNNs are a critical part of the AI landscape. For many years, topologies such as LSTM/GRU have dominated. However, these come with drawbacks – they often require a large number of parameters, they are not structured on any transforms (ad-hoc learning) and often require higher resolution activation (Eg 16bit).

LMU class of RNN is appealing in that it provides an alternative to the LSTM/GRU classes that are already established. I feel that it is important that this class of NN will be given exposure in the tinyML community.

I personally feel that for specific case of Keyword Spotting, there are other alternatives that should be compared (for example , Temporal Convolution). The main reason that I bring this up is because Keyword Spotting deals with a word spoken over a relatively short and known window of time (eg 0.25s to 0.75s). Because of this property, it is not necessary (though still possible) to use an RNN. Temporal convolution is capable of achieving great results for this sort of application with few parameters.

Having said that, there are plenty of applications where time series analysis/regression cannot be performed by a TCN (eg in cases where time window is not entirely known or arbitarily large). In those cases, there is still a critical need to identify an efficient method for performing this analysis. It is for this very reason that I believe that this work should be noted by the community.

For this reason, I strongly recommend that this paper be accepted to tinyML Research Symposium Summit.





**Paper Strengths:**

Recurrent Neural Networks are a very useful class of NNs. They are particularly useful in analysis of time-series data where durations of events are not fixed and thus are not malleable to analysis via fixed-dimension topology NNs such as CNN. The traditional RNNs (eg LSTM), however, are parameter intensive, allow no weight re-use, are more complex structurally and are not very explainable as their state advancing is learnt in training and is not based on any specific transforms understandable by humans.
LMU is a relatively new idea which introduces a different class of RNN, one which is based on using Legendre basis for compressing information over a time window.
The result is a simpler RNN with fewer parameters which according to the authors outperforms the existing alternatives.
Furthermore, this paper presents a method to achieve a very low bit resolution (down to 4bit activation) RNN. This in itself may be an extremely useful advancement.



**Paper Weaknesses:**

A few questions on claims
-	the LMU outperforms the LSTM with a 10^6x reduction in error, while encoding 10^2 more timesteps – this does not sound right – unless it’s not clearly stated
-	Power numbers for CortexM4F are questionable because actual real-life power comprises not only from compute resource, but perhaps even more dominated by movement of data to and from memories
-	Furthermore, a certain portion of the power needs to be accounted for in data-acquistion (eg ADC whether it be via Sigma-Delta microphone and requires a decimation filter, or an Analogue microphone which requires an onboard ADC.

It would be good to understand how LMU method would compare against a Temporal Convolution method.


**Poster (If Paper Is Rejected):**

1: Yes, ok for poster sesion to nurture work

**Reviewer Confidence:**

5: The reviewer is absolutely certain that the evaluation is correct and very familiar with the relevant literature

---

### Official Review · AnonReviewer3 · 2021-01-30

**Overall Merit Score:** 2

**Brief Summary:**

In this paper, hardware-aware training has been employed to build new neural networks based on the Legendre memory unit (LMU) for keyword spotting applications. The previously proposed LMU algorithm has been adapted in several ways to make it more efficient for hardware and improve performance.
Algorithm results for Google’s KWS dataset have been compared against state-of-the-art works, and power consumption based on custom RTL design has been reported.

**Detailed Comments:**

The authors did not abide by the double-blind policy, since it has been stated in Section 2 that [13] is the authors’ previous work.

For the hardware design and evaluation, it is strongly suggested that the authors should go through the proper RTL synthesis and place-and-route tool flows, to complete a real custom hardware design and report the simulated power consumption of the actual hardware (instead of crudely ‘estimating’ the power based on the ‘estimated’ number of transistors).

In Section 4.2, the authors criticized other works that can only detect a small of keywords but did not clearly report how many keywords the authors’ proposed work here can detect. This should be clarified.

In Fig. 3, it is not clear why there is a sweet spot around 8M transistors and why the power consumption is lowest around that region. If the number of transistors reduced below 5M transistors, why do the clock frequency and power consumption increase steeply?


**Paper Strengths:**

The authors modified the previously proposed LMU algorithm to make it more hardware-efficient without sacrificing accuracy. While the keyword spotting problem has been studied extensively by many prior works, this work still pushes the boundary of attainable accuracy with the smallest model size (in the number of total bits).

**Paper Weaknesses:**

The hardware design and evaluation are not comprehensive. It seems the authors estimated the number of transistors for the proposed custom hardware design based on [16], and then the dynamic power is estimated based on the number of transistors, so there are two levels of estimation here. It is not clear whether the power estimation obtained by this kind of crude method is really credible at all.

**Poster (If Paper Is Rejected):**

1: Yes, ok for poster sesion to nurture work

**Reviewer Confidence:**

5: The reviewer is absolutely certain that the evaluation is correct and very familiar with the relevant literature

---

### Decision · Program_Chairs · 2021-02-05

**Decision:**

Accept (Regular)

**Comment:**

Congratulations on your paper's acceptance!

Your paper has been accepted as a full-length regular paper.

Please read the reviews carefully and make sure the concerns are addressed in your final submission.

All accepted papers will be given a slot in the TinyML Summit schedule for an oral presentation on Friday, March 26, 2021.

Camera ready instructions will follow soon. All papers will be hosted on arXiv and published papers will have the following header stamp: “Published as a conference paper at TinyML Research Symposium 2021.” The paper will also be presented on the program website.